# Characterization of Equilibrium Catalysts from the Fluid Catalytic Cracking Process of Atmospheric Residue

**Seybou Yacouba Zakariyaou [1,2]**, **Hua Ye [1,\***, **Abdoulaye Dan Makaou Oumarou [3]**, **Mamane Souley Abdoul Aziz [3]** and **Shixian Ke [1]**

[1] School of Chemistry and Chemical Engineering, Central South University, Changsha 410083, China; 212318002@csu.edu.cn (S.Y.Z.); ksx15172035644@163.com (S.K.)

[2] Département du Génie Pétrolier, Institut Universitaire de Technologie, Université André Salifou de Zinder, Zinder B.P.656, Niger

[3] Département des Energies Fossiles, Université d'Agadez (UAZ), Agadez B.P.199, Niger; dan.abdoulaye@yahoo.fr (A.D.M.O.); maman15abdoulaziz@yahoo.com (M.S.A.A.)

\* Correspondence: yehua308@csu.edu.cn

**Abstract:** In the FCC conversion of heavy petroleum fractions as atmospheric residues, the main challenge for refiners to achieve the quantity and quality of various commercial products depends essentially on the catalyst used in the process. A deep characterization of the catalyst at different steps of the process (fresh, regenerated, and spent catalyst) was investigated to study the catalyst's behavior including the physicochemical evolution, the deactivation factor, and kinetic–thermodynamic parameters. All samples were characterized using various spectroscopy methods such as $N_2$ adsorption–desorption, UV-visible spectroscopy, Raman spectroscopy, LECO carbon analysis, scanning electron microscopy (SEM), X-ray diffraction (XRD), X-ray fluorescence (XRF), nuclear magnetic resonance spectroscopy (NMR$^{13}$C) analysis, and thermogravimetric analysis. The results of the $N_2$ adsorption–desorption, UV-vis, Raman, LECO carbon, and SEM imaging showed that the main causes of catalyst deactivation and coking were the deposition of carbon species that covered the active sites and clogged the pores, and the attrition factor due to thermal conditions and poisonous metals. The XRD and XRF results showed the catalyst's physicochemical evolution during the process and the different interlinks between catalyst and feedstock (Nickel, Vanadium, Sulfur, and Iron) elements which should be responsible for the coking and catalyst attrition factor. It has been found that, in addition to the temperature, the residence time of the catalyst in the process also influences catalyst structure transformation. NMR$^{13}$C analysis revealed that polyaromatic hydrocarbon is the main component in the deposited coke of the spent catalyst. The pyridine-FTIR indicates that the catalyst thermal treatment has an influence on its Brønsted and Lewis acid sites and the distribution of the products. Thermogravimetric analysis showed that the order of catalyst mass loss was fresh > regenerated > spent catalyst due to the progressive losses of the hydroxyl bonds (OH) and the structure change along the catalyst thermal treatment. Moreover, the kinetic and thermodynamic parameters showed that all zones are non-spontaneous endothermic reactions.

**Keywords:** fluidized catalytic cracking; catalyst characterization; catalyst deactivation; coking mechanism; mass loss; catalyst kinetic; thermodynamic parameters





## 1. Introduction

Fluid Catalytic Cracking (FCC) is a conversion process in petroleum refining that can be applied to various feedstock ranging from gas oil to heavy crude oil [1,2]. The concept of catalytic cracking is similar to thermal cracking. Still, it differs in using a catalyst that is (theoretically) not consumed to improve process efficiency and product quality [3]. Currently, catalytic cracking is the most important conversion process in the refinery regarding the amount of feedstock treated and the type of zeolite catalysts used [4,5].

The conversion efficiency of the process depends on the physicochemical properties, deactivation factor, and coking mechanism of the catalyst used to crack a particular feedstock to achieve good conversion with more usable and environmentally friendly final products [6]. Moreover, the deactivation factor mainly depends on the coke deposited on the catalyst and the physical degradation of the catalyst [7,8], which are caused by metal compounds of the feedstock, polyaromatic–aliphatic components, and the process thermal conditions [9], including temperature, injection flow air, particle velocity, and catalyst-to-oil weight ratio [1,10,11].

In previous studies, J. Ihli et al. [12], Wallenstein et al. [13], and Liu et al. [14] suggested that the metallic impurities in the feedstock and the thermal conditions of the FCC process are responsible for the permanent deactivation of the catalyst due to the physicochemical changes. Israel Pala-Rosas et al. [15] suggested that the polyaromatic species covered the catalyst's surface, clogged its pores, and caused the deactivation of the catalyst by reducing its acidity. Haigang Zhang et al. [16] studied the effect of regeneration atmospheres (air and $O_2/CO_2$) on the structure transformation of the FCC catalyst (Kaolinite) and concluded that the microporosity of the catalyst increased by 32.4% due to the combustion damage to the pore structure.

Therefore, it is of great interest to refiners and catalyst manufacturers to understand in detail the physicochemical behavior of the catalyst, the catalyst components and the feedstock impurities cross-linkage, the deactivation factor, and the coking mechanism during the process [17–20]. And we noted that it should be interesting to study the current behavior of the catalyst towards non-hydrotreated atmospheric residues derived from crude oil of the *Agadem bloc* in Niger. This case has not yet been the subject of an in-depth study that takes into account the characteristics of the feedstock (in general case, the feedstock must be hydrotreated to remove undesired impurities in the form of VGO (vacuum gas oil)), including the residence time of the catalyst in the process as a new factor that may influence catalyst conversion and crystal-phase interlinkage during thermal treatment.

To carry out this work, industrial reaction and regeneration operating conditions were used to study the mechanism of catalyst deactivation, crystallography, and coke deposition. The tested fresh catalyst was compared with the regenerated and spent catalysts taken during the process. Several powerful methods were used to characterize the different tested samples including BET, UV-vis, Raman LECO, SEM, XRD, XRF, NMR[13]C, and TGA methods. A powerful mathematical model was developed using thermogravimetric analysis data to calculate the catalyst's kinetic–thermodynamic parameters at different thermal treatment stages.

Currently, researchers have generally referred to spectroscopy methods in their investigations into the deactivation and structural changes of catalysts [21,22]. For example, to date, solid-state nuclear magnetic resonance spectroscopy (NMR13C) is the only method recognized to be able to provide accurate information about different types of carbons in the catalyst coke deposit, such as aromatic and aliphatic carbons, etc. [22]. The LECO carbon analyzer with a solid-state infrared (IR) detector is a powerful tool that is still relevant in the industrial field for the estimation of carbon concentration in the catalyst [23]. The BET, XRF, and XRD are considered the best techniques for the characterization of the catalyst structure in terms of surface and porosity analysis, phase identification, and chemical bond interlinkage during the thermal process [24–26].

The main objective of this work is to provide a clear understanding of the physicochemical behavior of the catalyst and the evolution of catalyst deactivation during the industrial FCC process by characterizing and comparing the results of fresh, regenerated, and spent catalysts. We demonstrate the catalyst residence time to be one of the key parameters that influence the catalyst structure change during the FCC process. Moreover, this work provides in-depth studies on the thermal decomposition, and kinetic and thermodynamic parameters of the catalysts during the FCC process with atmospheric residues to understand the catalyst behavior well [27–29].

## 2. Results and Discussion

### 2.1. $N_2$ Isotherm and Pore Size Distribution Analysis

Figure 1 displays $N_2$ adsorption–desorption isotherms of the samples at $-195.781$ °C and the corresponding Barrett-Joyner-Halenda (BJH) pore size distribution based on the adsorption branch. Figure 2a shows that all the curves of the tested samples have similar features and represent a combination of type I and IV isotherms [30,31]. The adsorption branch of the curves, which has a type I isotherm, is related to the microporous structures of the catalyst [32]. The hysteresis loop appears above $p/p0 = 0.4$, for the fresh catalyst, which can be attributed to IUPAC H4-type, indicating a complex structure with mesopores and micropores [16,31]. The hysteresis loops for regenerated and spent catalysts above $p/p0 = 0.43$ can be attributed to the IUPAC H3-type, which at $p/p0 > 0.7$ indicates the absence of limiting adsorption due to the accumulation of coke over the mesopores of the catalyst [32].

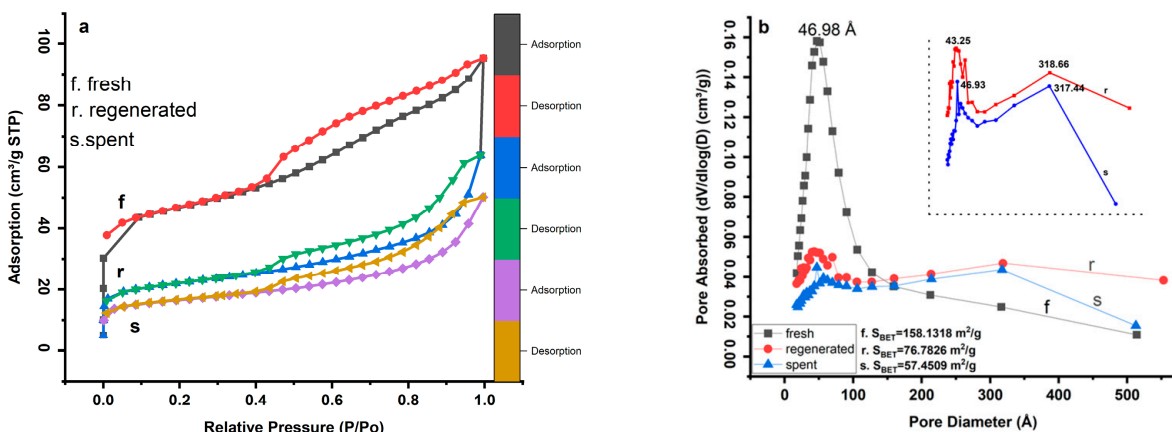

**Figure 1.** Brunauer-Emmett-Teller (BET) nitrogen adsorption and desorption isotherms (**a**) and BJH pore size distribution curves of tested samples (**b**): (f) fresh catalyst, (r) regenerated catalyst, and (s) spent catalyst.

At a relative pressure $p/p0$ of about 0, the sharp jump in the amount of $N_2$ adsorption, located on the fresh catalyst (curve (f)), gradually decreases on the regenerated catalyst (curve (r)) and finally disappears from the spent catalyst (curve (s)), indicating a progressive plugging of the micropore by the deposition of carbon species, as confirmed in Table 1. The gradual decrease in pore volume from fresh to spent catalyst is also evidence of pore plugging resulting from the deposition of carbon on the catalyst surface. The parallel features of the adsorption–desorption curves in the range of 0.55–0.75 and 0.53–75 for regenerated and spent catalysts, respectively, indicate that these large mesopores are open and may correspond to the catalyst particles [33].

**Table 1.** Textural properties derived from $N_2$ adsorption at 77 °K of the tested catalysts.

| Samples | $S_{BET}$ (m²g⁻¹) | $S_{ext}$ (m²g⁻¹) | $S_{micro}$ (m²g⁻¹) | $V_{total}$ (cm³g⁻¹) | $V_{micro}$ (cm³g⁻¹) | Microporosity (%) |
|---|---|---|---|---|---|---|
| Fresh | 158.13 | 94.24 | 63.8825 | 0.147 | 0.028 | 40.39 |
| Regenerated | 76.78 | 38.55 | 38.2327 | 0.098 | 0.017 | 49.79 |
| Spent | 57.45 | 28.87 | 28.5726 | 0.077 | 0.012 | 49.73 |

Microporosity = (Micropore Area/BET Surface Area) × 100.

Figure 1b displays the distribution of fresh, regenerated, and spent catalyst pore sizes. It can be observed that the fresh catalyst sample has one peak at 46.98 Å, while the regenerated and spent catalyst samples have two peaks at 43.25 Å–318.66 Å and 46.93 Å–317.44 Å, respectively. These second peaks at 318.66 Å and 317.44 Å indicate

the presence of mesopores in the samples due to the thermal conditions of reaction and regeneration [1]. As can be seen in Figure 1b, the uniform structure of the pore distribution of the fresh catalyst is significantly changed during the process. This process can be seen in the curves of the regenerated and spent samples, indicating a relatively large pore blockage in the pore diameter range from 159.98 Å to 318.66 Å.

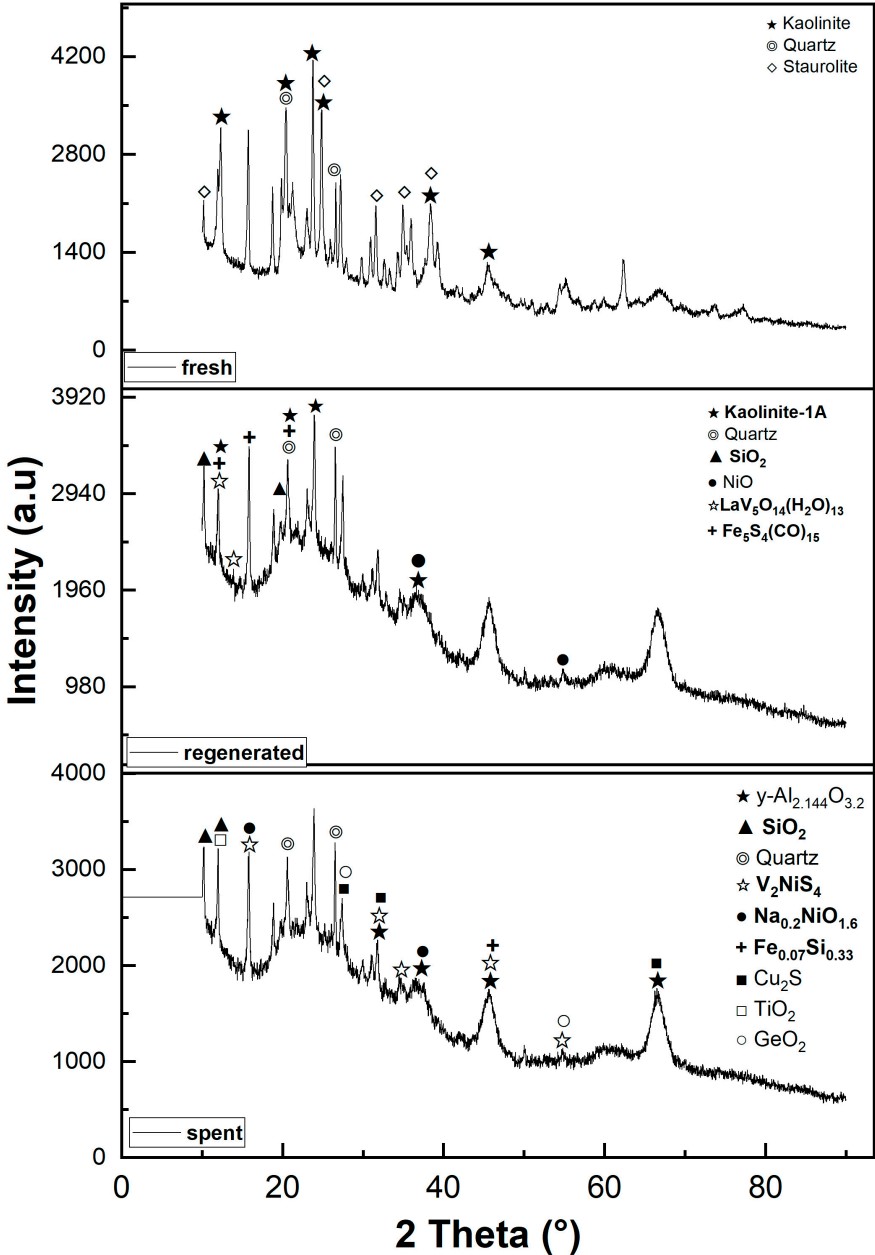

**Figure 2.** X-ray diffraction patterns of fresh, regenerated, and spent catalysts.

Table 1 shows that the BET surface area is greatly reduced during the process. The BET surface area of the fresh catalyst of 158.13 $m^2g^{-1}$ decreased by more than 50% compared to the regenerated catalyst of 76.78 $m^2g^{-1}$ and by more than 60% compared to the spent catalyst of 57.45 $m^2g^{-1}$. This event can be referred to the coke deposits on the catalyst surface and metal poisoning or the regeneration temperature above 690 °C, which changes the catalyst structure and porosity [34].

The calculated values of microporosity show that the microporosity of the regenerated catalyst (49.79%) is higher than that of the fresh and spent catalyst (40.39%) and (49.73%), respectively, which is due to the presence of a large mesopore peak of 318.66 Å created by

the effect of regeneration gas flow [1,24,35]. However, based on the values for the surface area and porosity of the regenerated catalyst compared to the values of the spent catalyst, it was found that there is a limit to the inverse relationship between surface area and porosity (the surface area of the catalyst decreases while its porosity increases) [36].

All the texture changes shown in Table 1, such as pore distribution, BET surface area change, microporosity, and pore volume, were directly related to the catalyst deactivation factor and coke mechanism, including the feedstock and the catalyst reaction-regeneration conditions [6,19,20,37].

## 2.2. Crystallography Analysis

The XRD patterns of all samples are shown in Figure 2. The XRD patterns of the fresh sample were compared with those of the regenerated and spent samples to better understand the evolution of catalyst structure due to thermal conditions during the process.

The alumina was present in the fresh sample in the form of $Al_2Si_2O_5(OH)_4$, kaolinite with a monoclinic system (ICDD: 04-026-7575) [38]. The characteristic peaks were found at $2\theta \approx 12.40°$, $21.23°$, $23.13°$, $24.96°$, $26.51°$, $38.36°$, and $45.40°$, corresponding to the reflections of the (001), (-1-11), (0-21), (002), (111), (-202), and (-203) planes. Other background peaks are associated with the crystalline phase of quartz (ICDD: 98-000-0369, [39]) with diffraction features at $2\theta$ values of $20.85°$ and $26.64°$ corresponding to the reflections of the (100) and (011) planes. In addition, the fresh XRD result confirms the presence of staurolite (ICDD: 98-000-0410); the regenerated catalyst showed that the peak intensities of kaolinite (kaolinite-1A) decreased due to the dealumination process regardless of the regeneration conditions. Kaolinite-1A (ICDD: 01-083-4643) [38] appeared essentially at $2\theta \approx 12.19°$, $20.04°$, and $24.53°$, corresponding to the reflections of the (001), (110), and (002)-planes, respectively. The silica amorph was also observed as one of the main crystalline phases in the regenerated sample, appearing at $2\theta \approx 10.13°$ and $19.81°$, corresponding to the reflections of the (010) and (013) (ICDD: 04-011-9355) [40] planes of $SiO_2$ species. The elements vanadium and nickel were clearly detected in the regenerated sample. The element vanadium is present in the form of lanthanum vanadium oxide hydrate with diffraction peaks at $2\theta \approx 12.10°$ and $13.83°$ (ICDD: 04-011-1951) [41], while the element nickel oxidized (NiO) with a diffraction peak at $2\theta \approx 37.76°$ and $54.48°$ (ICDD: 04-006-6545) [42]. These diffraction peaks correspond to the reflections of the (-111), (102) and (200), (220)-planes of $LaV_5O_{14}(H_2O)_{13}$ and NiO, respectively. It was also found that the sulfur reacts with the iron and other components to form iron-sulfur carbonyl. This $Fe_5S_4(CO)_{15}$ (ICDD: 04-010-2192) [43] appeared at $2\theta \approx 11.90°$, $15.98°$, and $20.89°$, corresponding to the reflections of the (102), (020), and (220) planes.

The XRD results show that the kaolinite-1A diffraction peaks become very weak and harder to detect in the spent catalyst. This may be related to the loss of the hydroxyl function OH, regardless of the process conditions and residence time of the spent catalyst. It appeared that the kaolinite-1A was converted to amorphous $y$-$Al_{2.144}O_{3.2}$ (ICDD: 98-000-0059) [44], silica (ICDD: 01-073-3406) [45], and residual kaolinite (kaolin serpentine, 1A) due to dihydroxylation and the presence of metal oxide species reacting with the kaolinite during the processing [46–48]. Qijun Hu et al. conducted a similar study on the evolution of the kaolinite structure and suggested that the temperature range 700–800 °C corresponds to the transition of meta-kaolinite into $\gamma$-alumina and silica [49]. The $y$-$Al_{2.144}O_{3.2}$ diffraction peaks appeared at $2\theta \approx 31.97°$, $37.68°$, $45.84°$, and $66.84°$, while the silica peaks appeared at $2\theta \approx 10.32°$ and $11.92°$. These peaks correspond to the reflections of the (220), (311), (440), (400), (440)-planes and (-113), (400), (-113)-planes of $y$-$Al_{2.144}O_{3.2}$ and silica, respectively. The remains of the kaolin-serpentine (ICDD: 04-025-7033) [50] appeared essentially at $2\theta \approx 12.36°$, $20.48°$, $21°$, and $24.87°$, corresponding to the reflections of the (010), (-101), (111), and (020)-planes. This is due to the fact that certain closed particles require a lot of energy to be dehydroxylated during thermal treatment at high temperatures. A large dispersion of vanadium, nickel, and sulfur was observed in the spent catalyst. In particular, the dispersion of vanadium peaks became more intense along the XRD curve. However,

due to the reaction with nickel and sulfur, the peaks were more visible than others and can be identified at $2\theta \approx 15.66$, $31.26$, $34.86°$, $45.50°$, and $54.81°$, corresponding to the reflections of (002), (110), (-112), (114), and (310) planes of $V_2NiS_4$ species (ICDD: 04-001-6617) [51]. In addition, the nickel was also associated with sodium to form sodium–nickel oxide, with diffraction reflections at $2\theta \approx 15.68°$ and $37.35°$, corresponding to the reflections of the (001) and (110) planes of $Na_{0.2}NiO_{1.8}$ species (ICDD: 04-009-2679) [52]. Three copper sulfide peaks were identified at $2\theta \approx 27.56°$, $31.93°$, and $66.78°$, corresponding to reflections of the (111), (1200), and (400) planes of $Cu_2S$ (ICDD: 03-065-2980 Barth, T. et al. (1926)). A diffraction peak that appeared at $2\theta \approx 45.62°$ is the characteristic peak of the iron silicon corresponding to the (110) plane of the $Fe_{0.07}Si_{0.33}$ species (ICDD: 04-002-3903) [53].

The crystalline phases of $GeO_2$ (ICDD: 04-021-5362 Li Y.F. et al. (2014)) and $TiO_2$ (ICDD: 98-000-0375) [54] were also detected at $2\theta \approx 11.76°$ and $27.44°$, $54.32°$, respectively. These peaks correspond to the reflections of the (211) and (110), (211) planes of $GeO_2$ and $TiO_2$, respectively.

The crystalline phases of Quartz stay unchanging at all steps of the process (regenerated and spent catalysts) and keep their diffraction peaks at $2\theta$ of $20.85°$ and $26.64°$.

According to the XRD analysis, during the thermal treatment of the catalyst, it was clearly observed that the texture changed due to the intermolecular reaction. This clearly indicates that some components lose their crystallinity and crosslink to form new structures, as shown in Figure 3A,B [1,16]. Under the same thermal conditions, there were no significant peaks of gamma alumina on the regeneration XRD curve in contrast to the spent catalyst. This occurrence of gamma alumina peaks in the spent catalyst shows that the residence time of the catalyst in the process has an important influence on the kaolinite transformation process. Furthermore, this means that at a constant thermal condition, the linkage and interaction of catalyst components depend on the residence time of the catalyst in the process. The longer the residence time, the deeper the reaction between the components.

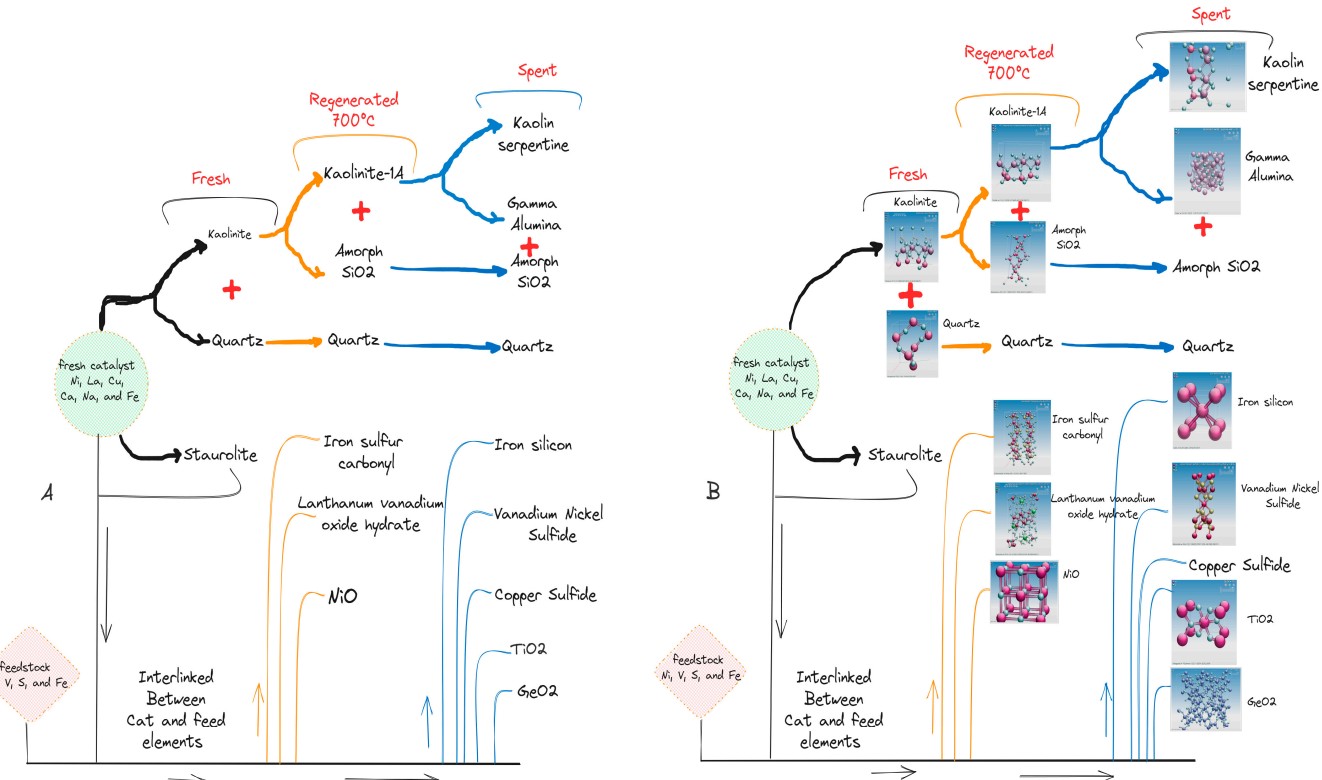

**Figure 3.** Scheme of catalyst components crosslinked and phases evolution (**A**,**B**) during the FCC process.

The presence of various forms of nickel, sulfur, and vanadium as major impurities in the regenerated and spent catalysts contributes to catalyst deactivation and attrition factors [7,8,22,55–57]. These poisons occur naturally in the non-hydrotreated atmospheric residue (feedstock) and are associated with various structures during the process. Marris D. et al. studied the effect of sulfur in the catalyst reaction and showed that Sulfur on the catalyst surface affects the performance of the catalyst activity [58]. U.J. Etim et al. concluded that vanadium and nickel have an important influence on the FCC catalyst behavior. The first one has an influence on the catalyst activity, while the second one favors coke formation through the mechanism of dehydrogenation reaction [8].

Table 2 presents the main tested sample's chemical oxide components, which were determined using a nondestructive analytical X-ray fluorescence technique to complete the XRD study. The XRF results show a gradual increase in iron content in catalyst samples during the catalytic cracking process of the atmospheric residual oil as feedstock (3.2 ppm of iron) [59], which may have some important implications, including a direct poisoning effect on the catalyst active sites, an effect on the selectivity of some reactions, a physical effect of iron deposition (pore plugging), and physical deconstruction [37]. The presence of lanthanum in the form of lanthanum oxide ($La_2O_3$) in the catalyst sample is related to improving catalyst resistance, assisting in the degradation of heavy molecular chains, and promoting good product stability [60–64]. In addition, the amount of alumina and silica, which are the main components of all tested samples, increased from fresh to regenerated catalyst, which can be attributed to the dealumination process [20].

**Table 2.** Composition of catalyst samples (wt.%) from XRF analysis.

| property | Fresh | Regenerated | Spent |
|---|---|---|---|
| | Composition (wt.%) | | |
| $CO_2$ | 9.64 | 3.56 | 9.58 |
| N | 0.27 | | |
| $Na_2O$ | 0.10 | 0.12 | 0.11 |
| MgO | 0.35 | 0.14 | 0.11 |
| $Al_2O_3$ | 50.71 | 50.28 | 47.04 |
| $SiO_2$ | 34.65 | 39.04 | 37.35 |
| $P_2O_5$ | 0.51 | 0.54 | 0.35 |
| $SO_3$ | 0.33 | 0.2 | 0.26 |
| $K_2O$ | 0.47 | 0.54 | 0.56 |
| CaO | 0.07 | 0.12 | 0.10 |
| $TiO_2$ | 0.22 | 0.19 | 0.20 |
| $V_2O_5$ | | 0.06 | 0.05 |
| $Fe_2O_3$ | 0.44 | 0.55 | 0.50 |
| $Co_2O_3$ | | 0.12 | 0.08 |
| NiO | | 2.05 | 1.52 |
| CuO | | 0.004 | |
| ZnO | 0.003 | 0.013 | 0.008 |
| $Ga_2O_3$ | 0.012 | 0.012 | 0.010 |
| $GeO_2$ | | 0.004 | |
| $Rb_2O$ | 0.002 | 0.003 | |
| $Y_2O_3$ | 0.66 | 0.14 | 0.08 |
| $ZrO_2$ | 0.003 | 0.004 | |
| $Sb_2O_3$ | | 0.05 | 0.02 |
| $La_2O_3$ | 1.18 | 1.66 | 1.61 |
| $CeO_2$ | | 0.48 | 0.32 |

### 2.3. UV-Visible Near-Infrared Spectra and Raman Spectra Analysis

Figure 4 shows the Raman spectra of fresh (f), regenerated (r), and spent (s) catalysts. The fresh catalyst exhibited Raman bands of 339.14, 1363.69, 1594.94, and 2915.96 cm$^{-1}$, while the regenerated catalyst showed bands of 143.93, 394.55, 508.68, 633.35, 1368.14,

1568.47, 2473.42, and 3104.40 cm$^{-1}$. The spectra bands of 215.37, 267.81, 655.64, 1093.30, 2513.65, and 3102.48 cm$^{-1}$ were identified from the spent catalyst.

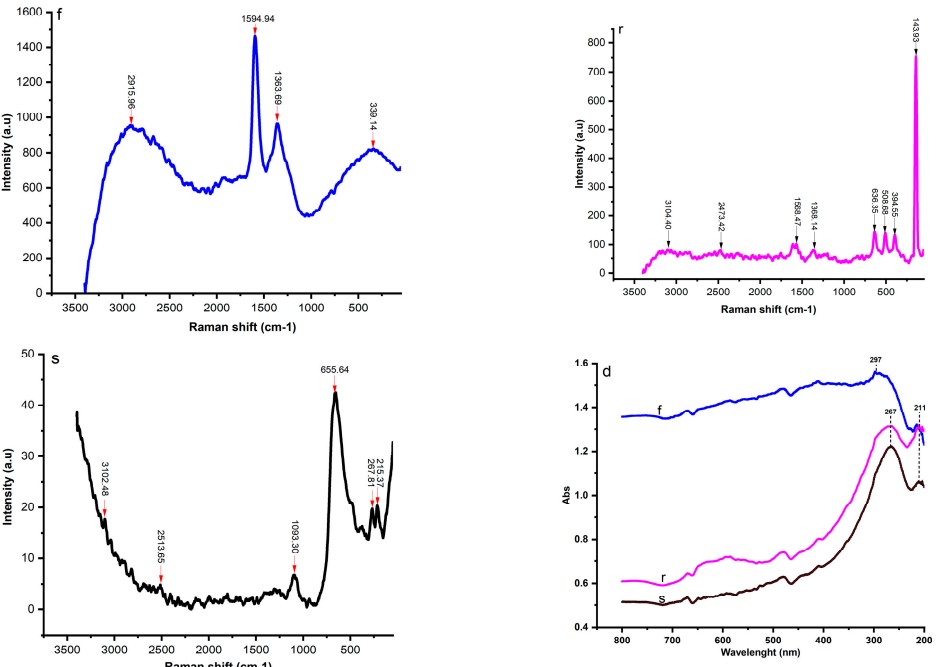

**Figure 4.** Raman spectra [fresh (**f**), regenerated (**r**), and spent catalyst (**s**)], and UV-Vis curves (**d**).

The bands in Raman spectrum of the fresh catalyst of 1363.69 and 2915.96 cm$^{-1}$ were identified and corresponded to the stretching vibration of bond P = 0 (P$_2$O$_5$) and bond OH, respectively, while the band of 339.14 cm$^{-1}$ was assigned to the bending mode of a 6-membered ring in the aluminosilicate zeolite structure [65].

The bands in Raman spectrum of the fresh regenerated catalyst of 143.93, 394.55, 508.68, and 636.35 cm$^{-1}$ can be assigned to the lattice vibrations in crystals (LA modes), the Si-O-Si bond, the asymmetric stretching vibration of GeS$_2$ and the stretching vibration of TiO$_2$ bond, respectively. The bands of 1368.47 cm$^{-1}$ and 1568.47 cm$^{-1}$ refer to the ring stretches of polyaromatic species and the ring stretches of Aliphatic azo, respectively. It is important to note that the bands in Raman spectrum of the regenerated catalyst of 1368.47 and 1568.47 cm$^{-1}$, indicative of the carbon structures, were located near the overlapping peaks of the D1 and G bands, which were a disordered graphite lattice (graphene layer edges) and ideal graphite lattice [16,66,67].

The bands of 2473.47 cm$^{-1}$ and 3104.40 cm$^{-1}$ of the regenerated catalyst corresponded to P-H and 0-H bonds, respectively. Interestingly, all the bands disappeared in the fresh and regenerated catalyst or became very weak in the spent catalysts due to the higher carbon content. However, the Raman spectrum of the spent catalyst showed a very low-frequency vibration that can be associated with the final stages of the crystallization process, such as the bonds at 215.37 and 267.81 cm$^{-1}$. The stretching vibration of the P=S bond at 655.64 cm$^{-1}$ was also identified as the main peak in the Raman spectra of the spent catalyst.

Figure 4d displays the UV-visible near-infrared spectra of fresh, regenerated, and spent catalysts. All curves have similar features and exhibit two important peaks between 214 nm and 297 nm. The UV-vis band of 214 nm is presented on both curves of the tested samples with low absorbance in the spent catalyst, which is due to the reduction in the Al content during the process and is assigned to the πp-πd transition between O$^{2-}$ and Al$^{3+}$ [47]. The band of the fresh UV-Vis spectrum of 297 $\lambda$max (nm) is related to the band of the regenerated and spent catalyst of 267 $\lambda_{max}$ (nm) and can be attributed to the C=O bond (π→π*).

### 2.4. Solid-State LECO Carbon Analyzer and NMR $^{13}$C Studies

A LECO carbon analyzer with a solid-state infrared detector (IR) was used to confirm and complete the $N_2$ adsorption–desorption, Raman, and UV-Vis results [23]. The test results indicated a carbon content of 0.5 wt.% in the regenerated sample and 1.26 wt.% in the spent catalyst. These values confirmed that the main cause of the catalyst deactivation factor was related to the carbon species deposited during the process.

Figure 5 shows the nuclear magnetic resonance spectroscopy (NMR$^{13}$C) spectra in the solid state, which helped us to learn the status of the deposited coke in the spent catalyst and to complete the above spectroscopy results. Only one strong resonance peak was observed at 125.24 ppm, which can be attributed to aromatic (100–150 ppm) carbons, indicating polyaromatic hydrocarbons in particular, regardless of the reaction-regeneration operating conditions [21].

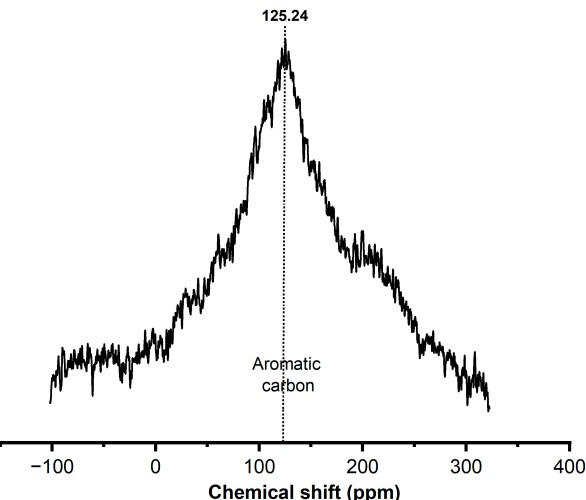

**Figure 5.** Nuclear Magnetic Resonance spectra of spent catalysts.

### 2.5. Catalyst Morphology Changes during the FCC Process

The images of crystal morphology of the tested fresh, regenerated, and spent catalyst samples with sizes of 200 μm and 50 μm are shown in Figure 6 using field emission scanning electron microscopy (SEM). The images of the fresh catalyst SEM shown in Figure 5a indicate that the particles were uniformly spherically charged and had a clean surface and boundaries. It was found that there were smaller spherical particles attached to the largest particle surfaces of the fresh catalyst, which may have happened during the preparation of the catalyst. As can be seen in the images of the fresh catalyst, the lightest area of the particles has crystallites, while the darkest area is support.

The regenerated SEM images (Figure 6b) show a significant physical texture change, the mesoporous structure almost disappears, and the particles become dark. This could be related to the metal poison [37,55] and coke deposited on the catalyst (0.04–0.05 wt.%), which was confirmed using the LECO carbon analysis. It can also be observed that the small particles attached to the large particles of the fresh sample were broken during the process and the shape of the particles became more surrounded due to the process temperature and running time, as reported by Xinzhuang Zhang et al. [68,69].

Figure 6c shows the images of the spent catalyst SEM after deactivation with a carbon content of 1.26 wt.% on the catalyst. These images show a complete disappearance of the mesopores, physical abrasion of the particles, and fragmentation into smaller pieces. These phenomena may relate to collisions between particles or particles against the tube [8], lateral cracks on the particle surface, and surface attrition due to stresses caused by fluid dynamics [9], as well as poisonous metals such as iron, nickel, vanadium, and sulfur, or the reaction between iron and calcium [57]. The catalyst's attrition such as catalyst abrasion and fragmentation, is an important factor affecting particle properties [7,69–71].

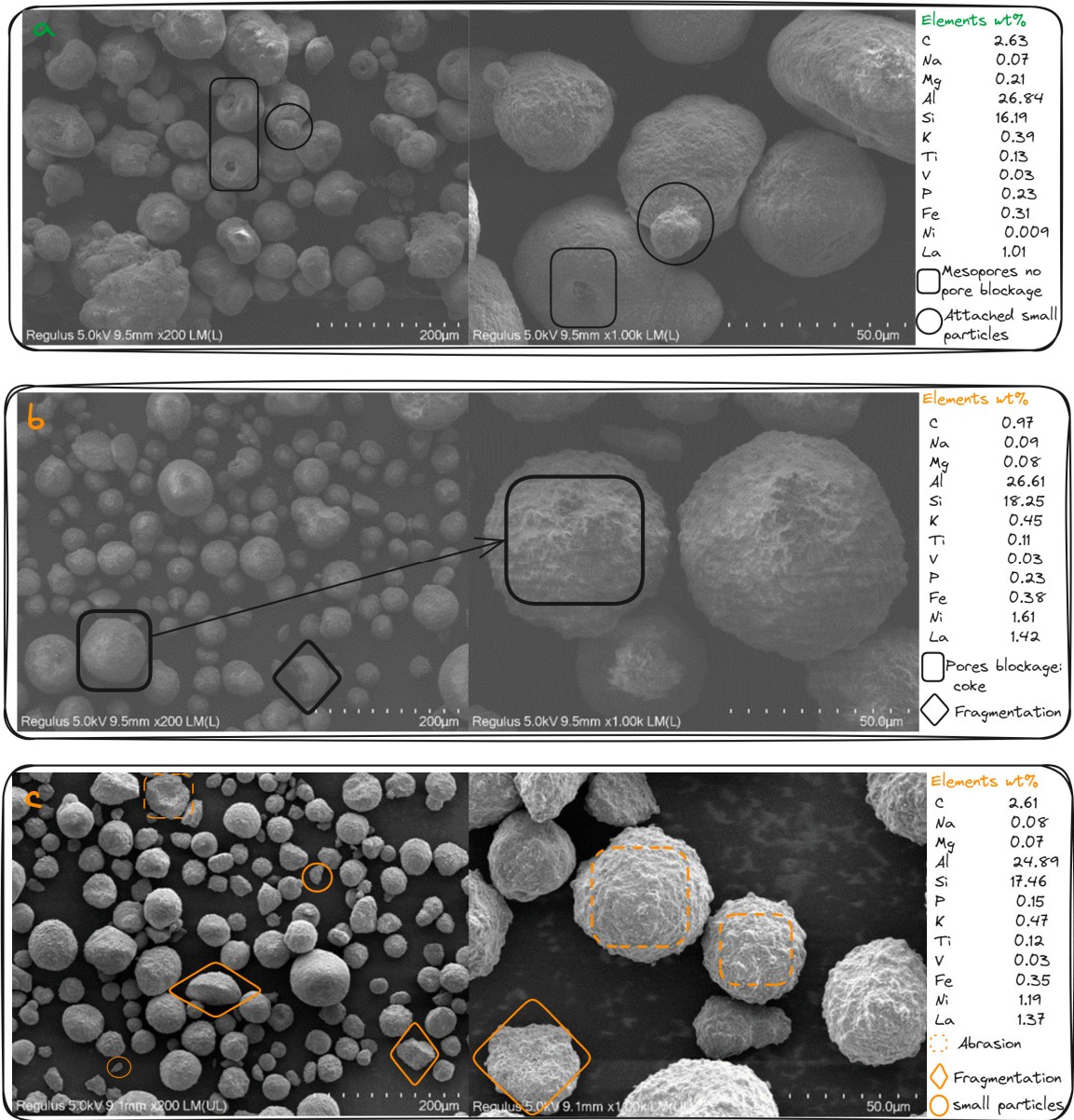

**Figure 6.** SEM images of the samples of fresh (**a**), regenerated (**b**), and spent (**c**) catalysts.

The profiles of the weight percentages of the main chemical elements confirm the thermal behavior of the catalyst components during the catalytic fluidized bed cracking and give a clear picture of it. From the Si/Al ratio shown in the Figures of SEM (a, b, and c), it can be observed that the regenerated catalyst cannot be recovered as fresh, and the more the catalyst is treated, the more the Si/Al ratio approaches 1, which is characteristic of spent catalyst [72].

### 2.6. Catalyst Acidity Evolution during the FCC Process

Figure 7a,b shows the pyridine FTIR profile of the fresh (f), regenerated (r), and spent (s) catalysts. This powerful method was used to study the evolution of the Brønsted and Lewis acid sites of the catalyst in the different stages of the FCC process. The absorption bands at 1540 and 1450 cm$^{-1}$ were identified as Brønsted and Lewis acid sites, respectively, while the band at 1490 cm$^{-1}$ was considered to be a combination of Brønsted and Lewis acid sites. The calculated concentrations of the Brønsted and Lewis acid sites are shown in Table 3. As far as the concentration of Brønsted and Lewis acid sites of the fresh catalyst

is concerned, a reduction of 71.92% and 96.37% in Brønsted acid sites and a reduction of 63.63% and 91.54% in Lewis acid sites were observed in the regenerated catalyst at 150 and 500 °C, respectively. The spent catalyst definitely lost its Brønsted acid sites with only 2 μmol g$^{-1}$ of Lewis acid sites at 500 °C. The concentration of the Brønsted and Lewis acid sites of the catalyst decreased from fresh to spent catalyst as a result of thermal treatment. The literature repeatedly points out that the Brønsted and Lewis acid sites influence the distribution of outlet products.

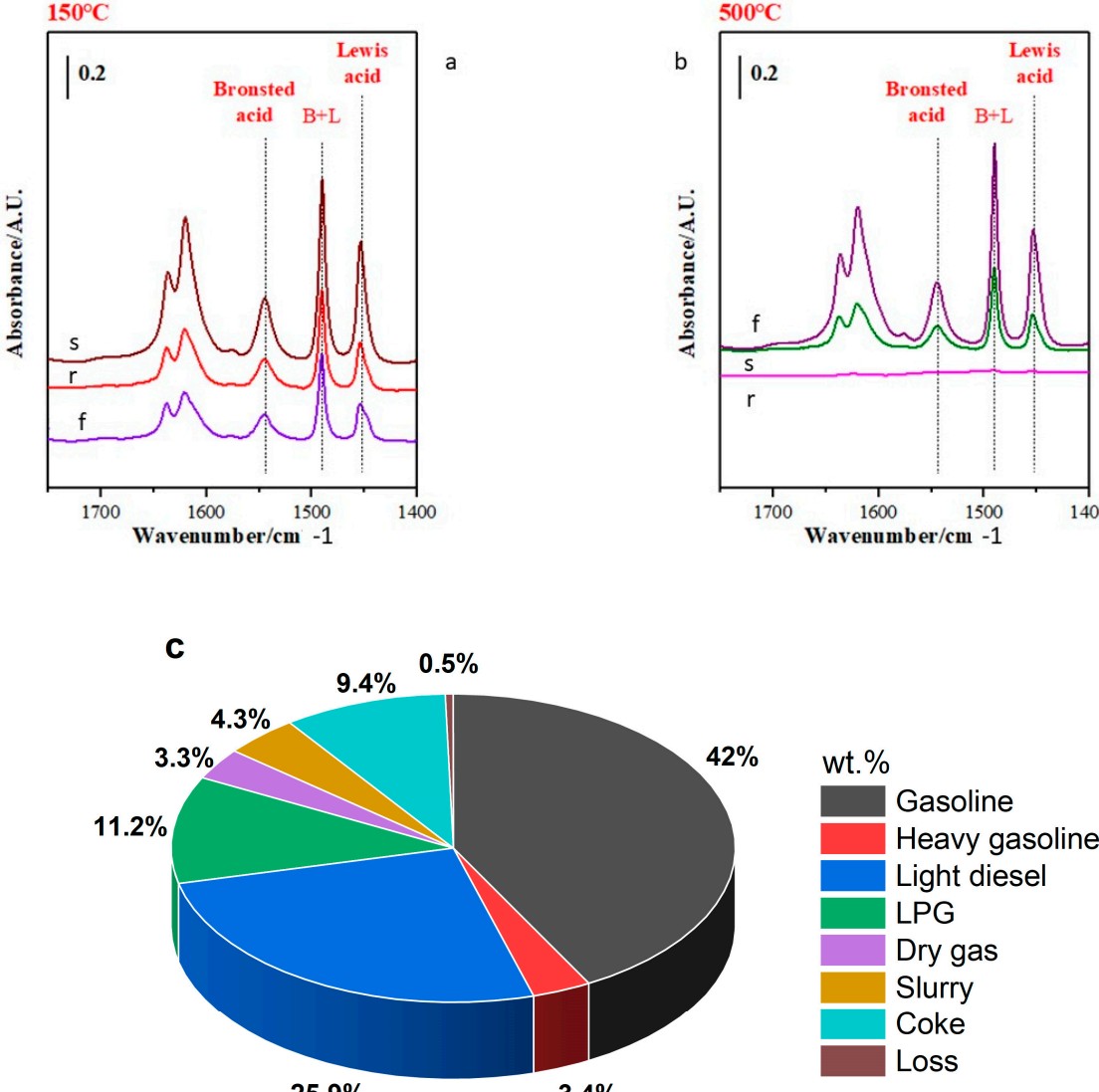

**Figure 7.** Pyridine adsorbed-FTIR spectra (**a**,**b**) [fresh (f), regenerated (r), and spent (s) catalysts] and distribution process's outlet products (**c**).

**Table 3.** Brønsted and Lewis acid site concentration at 150 and 500 °C.

| Samples | Pyridine, μmol g$^{-1}$ | | | |
|---|---|---|---|---|
| | **Brönsted Acid** | | **Lewis Acid** | |
| | **150 °C** | **500 °C** | **150 °C** | **500 °C** |
| Fresh | 89 | 138 | 121 | 71 |
| Regenerated | 25 | 5 | 44 | 6 |
| Spent | 14 | 0 | 37 | 2 |

The combination of both acid sites ensures a good conversion of the feedstock, while on the other hand, the Brønsted acid sites are responsible for the conversion of C-C bonds into light petroleum fractions [73] and the Lewis acid sites are responsible for coke production [74] due to their dehydrogenation reactions (hydrogen transfer). Figure 7c shows the product distribution of this process under the same conditions as described at the beginning.

As shown by the profile of the acid sites of the fresh and regenerated catalyst, the distribution of outlet products and the high conversion rate of diesel may be related to the Brønsted acid sites of the regenerated catalyst, which were very low to promote cleavage of the C-C bonds. Although the Lewis acid sites promoted coke formation, it is important to note that the limitation of the coke rate is sometimes due to the technology of the equipment used. For more details, the outlet products produced in this process, such as gasoline, diesel, LPG, coke, and dry gas, have been described in our previous work [59].

### 2.7. Thermal Analysis

The sample's thermal events were analyzed through a thermo-gravimetric (TGA) profile along with a derivative thermo-gravimetric (DTG). Figure 8 shows the behavior of uncalcined catalysts and their conversion curves from fresh, regenerated, and spent catalysts. The TGA results show that five main thermal events were identified as mass loss zones across all curves. For all samples, the temperature range RT (room temperature) to 100 °C was not considered because of the influence of moisture.

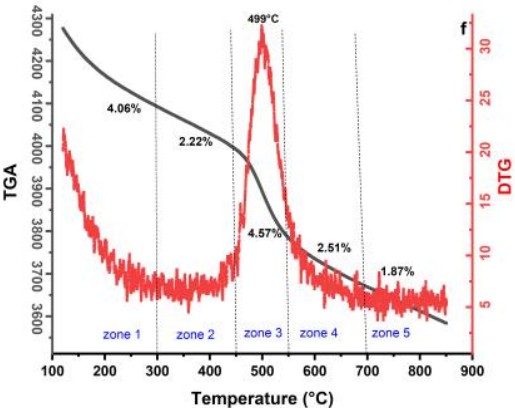
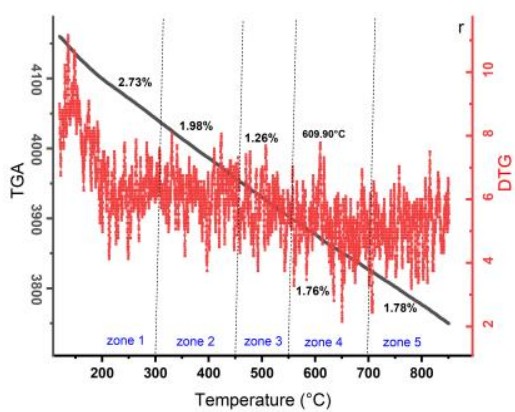

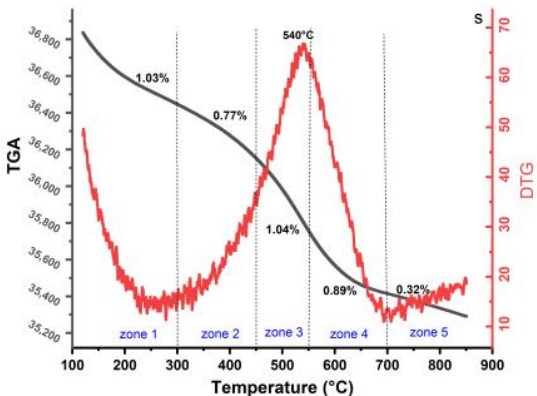

**Figure 8.** Mass loss for each zone: fresh catalyst (**f**), regenerated catalyst (**r**), and spent catalyst (**s**).

The first mass loss (zone 1) was measured in the temperature range of 100–300 °C. In this range, it is observed that the main mass losses for fresh, regenerated, and spent catalyst samples are estimated to be 4.06 wt.%, 2.73 wt.%, and 1.03 wt.%, respectively. This

first event can be attributed to the release of water bonds from the tested catalysts as a dehydration phase on the surface of $SiO_2$ in the tested samples. In this step, it is interesting to note that the mass loss values indicate a gradual decrease from fresh to spent catalyst, which is due to the chemical change in the catalyst during the process. This process can be explained by the fact that the density of hydroxyl groups (OH) is higher in the fresh catalyst and decreases during the thermal treatment [69,75].

For all zones, the mass losses of the fresh catalyst are associated with the progressive removal of its components, including the removal of co-precipitated nitrate, NiO, $GeO_2$, $Rb_2O$, and $ZrO_2$ over the temperature zone ranges. However, in the process reaction that occurred at a temperature of 500 °C, the mass loss profiles and the endothermic DTG peak at 499 °C showed that the fresh catalyst lost significant weight before the reaction took place in the riser–reactor.

The second and third mass loss in the regenerated and spent samples occurred between 300 °C and 550 °C and was the most important phase due to the combustion of coke on the catalyst surfaces.

In the third zone, the DTG curves of the fresh and spent catalysts exhibited strong endothermic peaks at 499 °C and 540 °C, respectively. These peaks can be attributed to the decomposition phase, i.e., the penetration of heat between the layers of decomposed materials, which led to the decomposition of hydrocarbons.

The fourth zone was particularly characterized by the fine and small exothermic peak at 609.90 °C from the regenerated DTG curve. Possibly, this peak is related to the inert carbon or graphitic carbon in the catalysts, as confirmed in the first part of the Raman spectra analysis.

The mass loss in zone 4 (600–700 °C) and above for all samples may be related to the oxidation of the metallic components and crystalline alteration of the catalyst supports, as reported in several studies in the literature [76–78].

The equilibrium catalyst behavior at different stages of the FCC process (fresh, regenerated, and spent catalyst) showed a significant change in its composition, which can be seen from the mass loss profiles presented in Table 4.

Accordingly, in the fifth mass loss zone 1.87 wt.%, 1.78 wt.%, and 0.32 wt.% for fresh, regenerated, and spent catalysts, respectively, there were no other significant weight losses up to 850 °C. This event could be referred to the final hydrocarbon oxidation and the conversion of the trapped metal into metal oxides [76].

**Table 4.** Kinetic and thermodynamic parameters of each zone of tested catalyst samples.

| | | Fresh Catalyst | | | | | | Regenerated Catalyst | | | | | | Spent Catalyst | | | | | |
|---|---|---|---|---|---|---|---|---|---|---|---|---|---|---|---|---|---|---|---|
| Zone | T °C | $Ea$ Jmol$^{-1}$ ($\times 10^2$) | $\Delta S$ Jmol$^{-1}$ ($\times 10^{-2}$) | $\Delta H$ Jmol$^{-1}$K$^{-1}$ ($\times 10^5$) | $\Delta G$ Jmol$^{-1}$ ($\times 10^6$) | Fitting Equation | $R^2$ | $Ea$ Jmol$^{-1}$ ($\times 10^2$) | $\Delta S$ Jmol$^{-1}$ ($\times 10^{-2}$) | $\Delta H$ Jmol$^{-1}$K$^{-1}$ ($\times 10^5$) | $\Delta G$ Jmol$^{-1}$ ($\times 10^6$) | Fitting Equation | $R^2$ | $Ea$ Jmol$^{-1}$ ($\times 10^2$) | $\Delta S$ Jmol$^{-1}$ ($\times 10^{-2}$) | $\Delta H$ Jmol$^{-1}$K$^{-1}$ ($\times 10^5$) | $\Delta G$ Jmol$^{-1}$ ($\times 10^6$) | Fitting Equation | $R^2$ |
| 1 | 100–300 | 10.85 | −3.03 | 51.73 | 5.34 | Y = −0.13065x − 0.0994 | 0.99 | 12.37 | −3.02 | 58.93 | 6.06 | Y = −0.1488x − 0.0984 | 0.99 | 42.45 | −2.86 | 202.24 | 20.38 | Y = −0.5106x + 0.5913 | 0.98 |
| 2 | 300–450 | 21.44 | −3.00 | 128.89 | 13.10 | Y = −0.2579x + 0.318 | 0.98 | 31.01 | −2.94 | 186.42 | 18.85 | Y = −0.373x + 0.6749 | 0.99 | 66.63 | −2.84 | 400.58 | 40.26 | Y = −0.8015x + 1.0601 | 0.97 |
| 3 | 450–550 | 133.44 | −2.63 | 913.11 | 91.52 | Y = −1.605x + 3.2677 | 0.98 | 52.08 | −2.88 | 356.37 | 35.87 | Y = −0.6264x + 1.252 | 0.99 | 201.00 | −2.59 | 1380 | 138 | Y = −2.4176x + 3.296 | 0.98 |
| 4 | 550–700 | 77.43 | −2.82 | 626.40 | 62.91 | Y = −0.9313x + 2.0443 | 0.99 | 79.01 | −2.84 | 639.18 | 64.19 | Y = −0.9503x + 1.8307 | 0.99 | 190.16 | −2.65 | 1540 | 154 | Y = −2.2872x + 3.2081 | 0.97 |
| 5 | 700–850 | 137.75 | −2.72 | 1290 | 129 | Y = −1.6568x + 3.0534 | 0.98 | 164.49 | −2.69 | 1540 | 154 | Y = −1.9784x + 3.275 | 0.98 | 134.56 | −2.77 | 1260 | 126 | Y = −1.6185x + 2.4638 | 0.98 |

## 2.8. Kinetics and Thermodynamics Parameters Calculation Using TGA

Assuming that all zone conversions are first-order reactions, the model of Coats and Redfern was developed to calculate the kinetic and thermodynamic parameters for each zone using the following equation and the constants of Plank and Boltzmann [79–82]:

$$\ln[-\ln(1-x)] = ln\frac{ART^2}{\beta E_a} - \frac{E_a}{RT} \text{ Where} x = \frac{wi - wt}{wi - wf}$$

$wi$: initial weight, $wt$: weight of sample at particular temperature $T$, $wf$: final weight.

$A$: pre-exponential factor, $\beta$: heating rate (10 °C/min), $R$: gas constant (8.3143 Jmol$^{-1}$K$^{-1}$), $Ea$: activation energy, $T$: temperature (K) at the peak of the DTG curves.

The slope and intercept values of each zone were obtained by plotting (Figure 9a) $\ln[-\ln(1-x)]$ versus $1000/T$, then we used these values to calculate apparent activation energy and other thermodynamic parameters listed in Table 4, such as pre-exponential factor $(A)$, Gibbs free energy $(\Delta G)$, Entropy Changes $(\Delta S)$, and Enthalpy changes $(\Delta H)$ [27,82–84].

$$E_a = slope \times 8.3143$$

$$A = \frac{\beta E_a e^{\frac{Ea}{RT^2}}}{RT^2}$$

$$\Delta S = R \times ln\left(\frac{Ah}{KT}\right)$$

$$\Delta H = E_a - RT$$

$$\Delta G = \Delta H - T.\Delta S$$

where $K$ is Boltzmann's constant (1.381 $\times$ 10$^{-23}$ J·K$^{-1}$), and $h$ is Planck's constant (6.626 $\times$ 10$^{-34}$ J s).

The obtained positive values of entropy changes $(\Delta S)$, enthalpy changes $(\Delta H)$, and negative values of $(\Delta S)$ proved that all zones were non-spontaneous endothermic reactions [85–89]. In zone 1, lower enthalpy values are observed in all the samples studied, which is due to physical adsorption due to hydrogen bonding and other complex reactions. The high enthalpy and energy values of zone 5 in the regenerated and spent catalysts imply that all the deposited coke on the catalysts in these temperature ranges was removed and the remaining structural bonds of the catalyst require more energy to be broken.

Figure 9b–d shows the five-point zones of all samples on the curves of activation energy and conversion with temperature and kinetic function. It was demonstrated that the activation energies change from each conversion zone. This behavior shows the complexity of the thermal decomposition due to the progressive degradation of the catalyst structure, such as the removal of components like nitrogen, H$_2$O, and other components. It seems that the deeper the catalyst is treated, the more the catalyst loses the hydroxyl bonds (OH) and other degradation products, and the more energy is required [75]. The variation of activation energies in different zones is sometimes related to the H/C ratio in the sample [90]. The lower $Ea$ values in zone 1 are generally related to the breaking of hydroxyl bonds, and high energy is not required to break these bonds such as light components. The lower $Ea$ and higher conversion values (0.67) of the regenerated catalyst observed in zone 3 compared to the values of the fresh and spent catalyst may be related to the remaining soft-coke decomposition in this 450–550 °C range. The lower $Ea$ value observed in zone 5 (134.56 $\times$ 10$^2$ Jmol$^{-1}$) of the spent catalyst compared to its zone 4 (190.16 $\times$ 10$^2$ Jmol$^{-1}$) may relate to the temperature range of 700–850 °C to remove the hard coke in the sample with a significant mass loss (1.78 wt.%) [91]. As shown in Figure 9b, the maximum apparent activation energies are reached at a conversion of 0.95, 0.93, and 0.77 for the fresh, regener-

ated, and spent catalysts, respectively. The Arrhenius equation ($K = A.e^{-Ea/RT}$) was used to calculate the kinetic constant of each zone, as shown in Figure 9c. The kinetic curve of the spent catalyst showed little fluctuation at low $K$ values due to the highest $Ea$ values, indicating that the term oxidation rate decreased in the same way as the activation energy function. The regenerated curve also shows that in soft-coke decomposition (450–550 °C), the kinetics automatically increased with decreasing energy, indicating that the kinetic values are strongly dependent on the activation energy values. In this case study, the behavior of the kinetic rate in non-spontaneous endothermic reactions can be explained by the reactions requiring the absorption of energy, an increase in temperature will provide the reactant molecules with more energy. As a result, a larger fraction of the collisions will have enough energy to overcome the activation energy barrier.

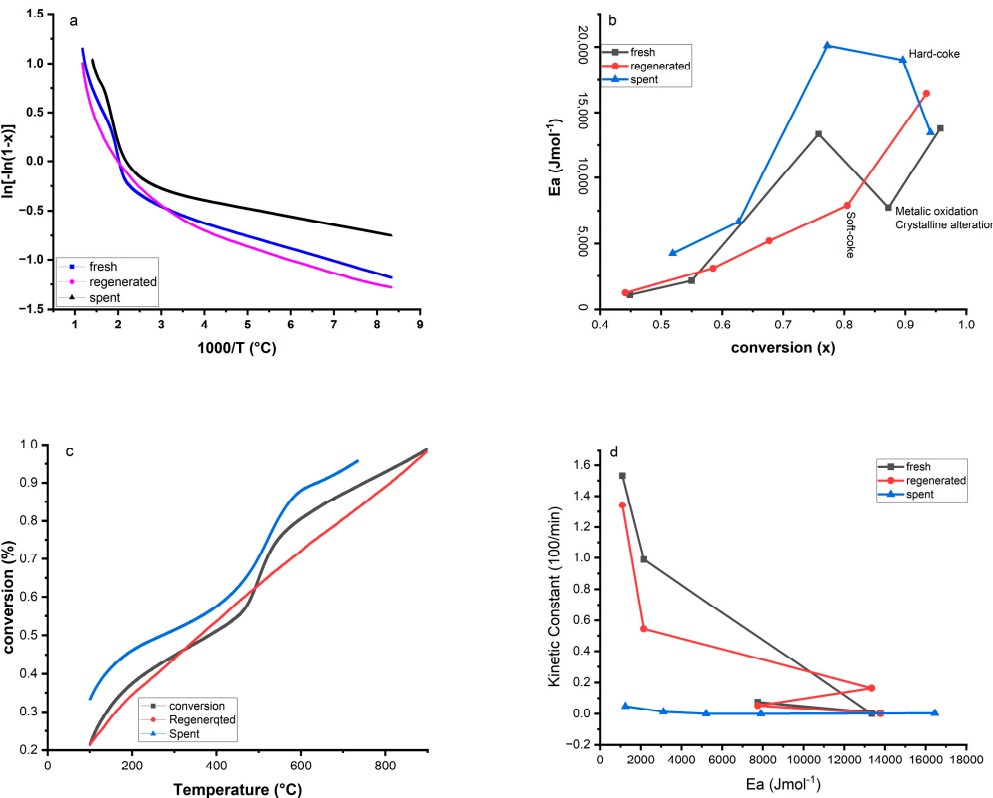

**Figure 9.** A Plot of $\ln[-\ln(1-x)]$ over $1000/T$ (**a**), apparent activation energy as a function of the coke concentration (**b**), and conversion as temperature function (**c**), and Kinetic constant coke therm-oxidation as a function of activation energy (**d**).

As shown in Table 4, all $R^2$ values are close to 1, indicating good agreement between the model and the experimental data used to estimate the kinetic parameter [28].

## 3. Material and Methods

### 3.1. Sampling of Catalyst

A non-hydrotreated atmospheric residue derived from Niger's crude oil of Agadem bloc was cracked over an equilibrium catalyst (Figure 10) in an industrial fluidized bed reactor unit of the Zinder Oil refining company in Niger. The spent and regenerated catalysts were sampled to conduct the experiments included in this paper.

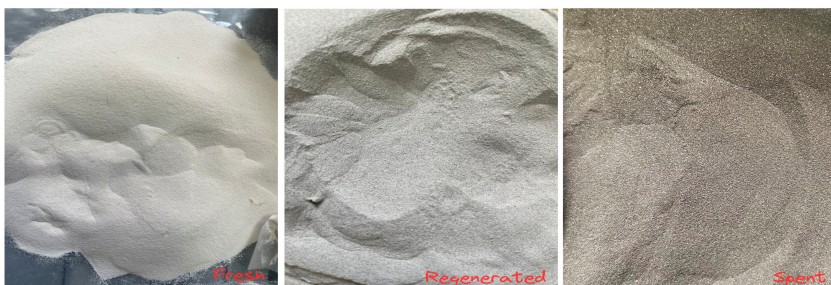

**Figure 10.** Physical appearance of fresh, regenerated, and spent catalysts.

### 3.2. Cracking Reaction and Regeneration Conditions

This study was conducted on a fluidized bed catalytic cracking unit of Zinder Oil Refining Company Limited in Niger, where the samples for this work were collected. The cracking reaction was carried out continuously using a non-hydrotreated atmospheric residue as feedstock, with properties listed in Table 5. The reaction was carried out at 500 °C in the range of 5–9 as a catalyst-to-oil weight ratio, and 3.3 s of oil gas residence time in the riser. The catalyst regeneration was performed at 700 °C as the regenerator bed temperature, under 0.26 MPa of regenerator top pressure, and a main air blower outlet flow of 1411 Nm$^3$/min (wet).

**Table 5.** Feedstock properties: non-hydrotreated atmospheric residue [59].

| Item | Value |
|---|---|
| Density (g/cm$^3$) | 0.9215 |
| API | 22.05 |
| Aniline point (°C) | 52.9 |
| Residual carbon (wt.%) | 5.5 |
| Nickel (ppm) | 22.9 |
| Vanadium (ppm) | 0.3 |
| Iron (ppm) | 3.2 |
| Sulfur (wt.%) | 0.24 |
| Nitrogen (wt.%) | 0.17 |
| Average molecular weight (g/mol) | 462 |
| Distillation TBP (°C) | |
| 50% | 343.0 |
| 90% | 378.0 |
| 95% | 385.0 |
| **SARA fractions (wt.%)** [a] | |
| Aromatics | 18.34 |
| Asphaltenes | 7.10 |
| Saturates | 33.44 |
| NS0 [b] | 41.12 |

[a] SARA analysis [37]; [b] NSO compounds (S, O, and N).

### 3.3. Catalyst Characterization

Various measurements were taken on all samples to determine catalyst morphology, structural properties, mass losses, chemical elements, and deposited coke.

N$_2$ adsorption–desorption isotherms at −195.781 °C were recorded using Micromeritics ASAP 2020 Unated-StatesV4.02 (V4.02 J). The specific area (S$_{BET}$), pore volume (PV), and pore size for both samples were measured using the Brunauer–Emmett–Teller (BET) method, depending on the isotherms of nitrogen adsorption/desorption. In contrast, the Barrett–Joyner–Halenda (BJH) method was used to determine the pore volume.

The crystal structures and phase composition evolution for all samples were investigated using X-ray powder diffraction (XRD) with an Empyrean Malvenpanako Sharp X-ray diffractometer (40 kV, 40 mA) using Cu *Kα* (*λ* = 1.54 Å) radiation, an angle range

(10–90°), and a rate of 10°/min. The diffraction reflections were assigned with the help of the International Centre for Diffraction Data (ICDD-PDF+2023—database). A model ZSX Primusll X-ray fluorescence (XRF) spectrometer (50 kV, 50 mA) was used to complete the XRD analysis and determine the samples' chemical elements and oxide components.

UV–visible near-infrared spectra were obtained on the Shimadzu UV 3600 Plus Spectrophotometer with Barium Sulfate Integrator configuration (ISR-603), scanning parameters for absorbance/transmittance in the range of 200–800 nm. Raman spectra were recorded on a DXR Raman spectrometer (Thermo Scientific, Waltham, MA USA, DXR2xi Micro-Raman) in the beam range of 50–3400 $cm^{-1}$ with scanning 900 times. The laser wavelength was 532 nm and the power was 2.4 MW.

Moreover, a LECO carbon analyzer with a solid-state infrared detector (IR) was used to confirm the presence and estimate the concentration of carbon in regenerated and spent catalysts in the oxygen environment.

The coke deposited on the spent catalyst's carbon type was determined using nuclear magnetic resonance ($NMR^{13}C$) spectroscopy. The spectra were recorded in a Bruker AVANCE III HD 600 MHz solid-state nuclear magnetic resonance spectrometer model operating at a frequency of 150 MHz using a 4 mm double-resonance solid probe at a rotor spinning rate of 10 kHz, high power decoupling sequence $^{13}C$ detection resonance, cycle delay time of 5 s, acquisition time of 32 ms, and scanning times of 1024 times.

The crystal morphology of all samples was observed using the field emission scanning electron microscopy (SEM) model Regulus 8100.

First, the Brønsted and Lewis acid sites of each sample were calculated using pyridine adsorption FTIR (model Nicolet iS50 spectrophotometer, Thermo Fisher Scientific, Waltham, MA USA) with a resolution of 4 $cm^{-1}$ and 256 scans. Secondly, 20 mg of each sample was used to prepare 14 mm diameter wafers under a pressure of 4 $t/cm^2$. The samples were calcined at 450 °C for 2 h and $10^{-3}$ Torr. The pyridine molecules were adsorbed at 100 °C and desorbed at 150 and 500 °C.

Thermogravimetric analysis (TGA) for all samples was performed via a model Japan Seiko TG/DTA7300 to determine the weight losses of the samples. This characterization was conducted in flowing air (100–150 mL/min) at a heating rate of 10 °C/min up to 900 °C.

The model of Coats and Redfern was developed to calculate the kinetic and thermodynamic parameters for each thermal event zone of all tested catalysts using the constants of Plank and Boltzmann.

## 4. Conclusions

In this work, different spectroscopy methods were investigated to characterize the equilibrium catalyst. During the cracking process in the fluidized catalyst, three major steps of the catalyst were performed, including the fresh state of the catalyst, after regeneration, and the spent state of the catalyst. According to the results of $N_2$ adsorption–desorption, Raman spectra, and UV-visible near-infrared spectra, as well as LECO carbon analysis, the catalyst deactivation and coking are attributed to the fact that the active sites of the catalyst were covered and the pores were blocked by the deposition of carbonaceous species during the process. However, the XRD analysis, supplemented by the XRF result, and the results from SEM show that the catalyst deactivation, coking, and attrition (particle abrasion and fragmentation) were caused by metal poisons (vanadium, nickel, and sulfur) of the feedstock, which reacted under the thermal conditions (500–700 °C) and under the influence of the catalyst residence time. It was also shown that the temperature of 700 °C is not sufficient for the initial decomposition of the kaolinite or meta-kaolinite to gamma alumina and amorphous silica, but that the reaction also requires time (residence time) to reach this transformation state. As already mentioned in the XRD analysis, the decomposition of the kaolinite under the same thermal conditions did not take place until the catalyst was spent. The crystallographic analysis showed that the deeper the catalyst is treated, the more the physicochemical properties change, and the crystal phase peaks

decrease due to the reaction and linkage between the phases, which consequently reduces the catalyst performance. The nuclear magnetic resonance spectroscopy (NMR$^{13}$C) analysis revealed that polyaromatic hydrocarbon is the main element as a coke compound in the spent catalyst, while Raman spectra indicated polyaromatic species, the ring stretches of Aliphatic azo, graphitic lattices, and ideal graphite lattices as carbon structures in the regenerated catalyst. The catalyst's acid sites evolution at different stages of FCC process indicated that the hydrothermal operating conditions influence the Brønsted and Lewis acid sites and the process product distribution.

The thermogravimetric analysis showed that the catalyst mass loss order is fresh catalyst > regenerated catalyst > spent catalyst due to the process's reaction regeneration, which affects the adsorption performance. In addition, the kinetic and thermodynamic parameters revealed that all zones are non-spontaneous endothermic reactions, and it was shown that the deeper the catalyst is treated, the more hydroxyl bonds (OH) are lost and the more activation energy is required. The apparent activation energy as a function of the coke conversion and the kinetic values changing from each zone indicate the complexity of the reaction during the catalyst's thermal treatment.

**Author Contributions:** Conceptualization, S.Y.Z. and H.Y.; methodology, S.Y.Z., A.D.M.O. and M.S.A.A.; software, S.Y.Z.; validation, H.Y.; formal analysis, S.Y.Z.; investigation, S.Y.Z.; resources, S.Y.Z. and H.Y.; writing—original draft preparation, S.Y.Z.; writing—review and editing, S.Y.Z., A.D.M.O., M.S.A.A. and S.K.; visualization, S.Y.Z., supervision, H.Y.; project administration, S.Y.Z. and H.Y.; funding acquisition, H.Y. All authors have read and agreed to the published version of the manuscript.

**Funding:** This research was funded by National Natural Science Foundation project 21975288 Preparation and Properties of two-dimensional Metal-Organic Framework (MOFs) Field-effect Materials (January 2020–December 2023).

**Data Availability Statement:** The data presented in this study are available on request from the authors.

**Acknowledgments:** The authors would like to thank Zinder Oil Refining Company (SORAZ) for supporting our studies.

**Conflicts of Interest:** The authors declare no conflict of interest.

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
