# Peer review of "Characterization of Equilibrium Catalysts from the Fluid Catalytic Cracking Process of Atmospheric Residue"

_catalysts, doi:10.3390/catal13121483_

Round 1
Reviewer 1 Report
Comments and Suggestions for Authors
The manuscript entitled Characterization of equilibrium catalysts from the fluid catalytic cracking process of atmospheric residue reports the detailed and exhaustive investigation of and characterization of the fluid catalytic cracking catalyst.
The authors conducted very relevant research. The authors use all possible methods to study the catalyst structure, morphology and thermal behavior of the catalyst. Based on the data obtained, the authors showed the peculiarities of thermal behavior of the studied catalyst. Using TGA data, the authors evaluated the kinetic and thermodynamic parameters of the catalyst evolution under the conditions of the cracking process.
The experimental evidence given is adequate and exhaustive. The conclusions drawn from the study are consistent with the evidence presented and meet the main purpose of the manuscript.
The manuscript has been written clearly and logically, and is well structured.
I think the article published will be interesting for the readers of the Journal and can be published in its current form. While, I would encourage the authors to revise and improve the language of the manuscript.
Comments on the Quality of English LanguageI believe that the language of the manuscript could be improved. The manuscript contains a number of stylistic errors. For example, there are phrases (lines 307, 311, 331) ...The fresh Raman bands ...The bands of the regenerated Raman spectrum ...The fresh bands. I suppose expressions like The bands in Raman spectrum of the fresh catalyst are more appropriate.
I recommend that you carefully check and improve the language of the manuscript.
Author Response
Dear Reviewer,
Hope this present letter finds you well.
We appreciate your precious time reviewing our paper and providing valuable comments and suggestions to improve this present work.
It was your valuable and insightful comments that led to possible improvements in the current version. The authors have carefully considered the comments and tried our best to incorporate changes to reflect most of the suggestions provided by you, point by point. You will find the graphical abstract (abstract section)
- In this current version add some prior work and main conclusions within the current literature. (Introduction section)
- We reinforce the section on the different techniques that are used to characterize the catalyst samples (found in the introduction section)
- We also reinforce some important information about the feedstock:
- Precision about the nature of the feedstock (non-hydrotreated atmospheric residue), the feed received from the distillation unit enters directly into the FCC unit without any hydrotreatment.
- Quality of English Language (Minor changes):
We would like to thank you for your suggestion about the English language. It will be improved, first by ourselves and improved also by the MDPI editing service before publication.
We sincerely appreciate all your valuable comments and suggestions, which helped us in improving the quality of the manuscript. We feel that your comments already have a positive effect on this paper, thank you for that. And we are available for you dear reviewer for any further suggestions or comments.
We will submit the last version before 16 November.
Reviewer 2 Report
Comments and Suggestions for Authors
This work aims to characterize an equilibrium catalyst for FCC process in 3 stages: fresh, spent and regenerated. Atmospheric residue is used as feedstock for cracking. The purpose of the work is of great interest for the scientific community. However, the methodology and results lack of novelty and on deepen on relating the effect of the kinetic performance with the catalyst deactivation and regeneration procedure.
- Introduction section does not mention any prior work and main conclusions within the current literature. This process is widely studied, and coke deactivation of FCC catalyst is a hot topic in this sense. They do not also reinforce the novelty of the feedstock, or catalyst, or new characterization techniques that are studied for this work. I recommend to thoroughly revise this section and to reinforce the novelty of the work, accordingly.
- Materials and methods section lacks of important information. The feedstock could be characterized more in detail. They do not mention if it is previously hydroteated. VGO is commonly used as feedstock and in this sense more information about the residue and the management of this feedstock prior to FCC unit shoul be mentioned.
-Information about reaction setup, product lumps and product analysis should also be mentioned, as well as extend the motivation for the operating conditions used.
- The catalyst used is not well described. No information about if it is agglomerated, fresh properties including acidity (TPD and/or FTIR analysis) should be included. Acid properties of the catalyst will be crucial for the cracking process.
- I suggest to include a kinetic results section, which will be later be correlated with all the characerization study of the catalyst in the 3 stages: fresh-spent-regenerated. Understanding the kinetic behavior of the catalyst will be crucial to understan which is the main origin of coke, etc.
Comments on the Quality of English LanguageMinor changes
Author Response
Dear Reviewer,
We hope this present letter finds you well.
We appreciate you for your precious time in reviewing our paper and providing valuable comments and suggestions to improve this present work.
Your valuable and insightful comments led to possible improvements in the current version. The authors have carefully considered the comments and tried our best to incorporate changes to reflect most of the suggestions provided by reviewers, point by point. We have highlighted the changes within the manuscript.
- You will find the graphical abstract (abstract section)
- In this current version add some prior work and main conclusions within the current literature. (Introduction section)
- We reinforce the section on the different techniques that are used to characterize the catalyst samples (found in the introduction section)
- We reinforce some important information about the feedstock:
- Precision about the nature of the feedstock (non-hydrotreated atmospheric residue), the feed received from the distillation unit enters directly into the FCC unit without any hydrotreatment.
- Including the feedstock compositions, SARA analysis (see Table 1)
- We included in the sampling section Figure (1) the Physical appearance of fresh, regenerated, and spent catalysts
- Your suggestion about the kinetic is fundamental. We improve this section by including the coke conversion, kinetic, and apparent activation energy curves.
- We really appreciate your attachment to improve this paper, thank you very much for that. Here, we also conducted Brønsted and Lewis acid sites using pyridine FT-IR, it included in before the last section with the process outlet products distribution (fig.8)
- Quality of English Language (Minor changes):
We want to thank you for your suggestion about the English language. It will be improved, first by ourselves and also by the MDPI editing service before publication.
Once again, we would like to express our gratitude for your insightful comments, which have undoubtedly contributed to improving our manuscript. We look forward to your continued support throughout this peer review process.
Reviewer 3 Report
Comments and Suggestions for Authors
This study presents very interesting data and investigation that deserves publication. I have some notes that have to be considered before publication:
1. Table 1: please add dimension of molecular weight
2. Table 3: Sentence “Metal composition” is not correct
Author Response
Dear Reviewer,
Hope this present letter finds you well.
We appreciate your precious time reviewing our paper and providing valuable comments and suggestions to improve this present work.
It was your valuable and insightful comments that led to possible improvements in the current version. The authors have carefully considered the comments and tried our best to incorporate changes to reflect most of the suggestions provided by you, point by point(Table 1 and 3). You will find the graphical abstract (abstract section)
- In this current version add some prior work and main conclusions within the current literature. (Introduction section)
- We reinforce the section on the different techniques that are used to characterize the catalyst samples (found in the introduction section)
- We also reinforce some important information about the feedstock:
- Precision about the nature of the feedstock (non-hydrotreated atmospheric residue), the feed received from the distillation unit enters directly into the FCC unit without any hydrotreatment.
- Quality of English Language (Minor changes):
We would like to thank you for your suggestion about the English language. It will be improved, first by ourselves and improved also by the MDPI editing service before publication.
We sincerely appreciate all your valuable comments and suggestions, which helped us in improving the quality of the manuscript. We feel that your comments already have a positive effect on this paper, thank you for that. And we are available for you dear reviewer for any further suggestions or comments.
We will submit the last version before 16 November.
Round 2
Reviewer 1 Report
Comments and Suggestions for Authors
The manuscript entitled Characterization of equilibrium catalysts from the fluid catalytic cracking process of atmospheric residue reports the detailed and exhaustive investigation of and characterization of the fluid catalytic cracking catalyst.
The authors conducted very relevant research. The authors use all possible methods to study the catalyst structure, morphology and thermal behavior of the catalyst. Based on the data obtained, the authors showed the peculiarities of thermal behavior of the studied catalyst. Using TGA data, the authors evaluated the kinetic and thermodynamic parameters of the catalyst evolution under the conditions of the cracking process.
The experimental evidence given is adequate and exhaustive. The conclusions drawn from the study are consistent with the evidence presented and meet the main purpose of the manuscript.
The manuscript has beeт written clearly and logically, and is well structured.
I think the article published will be interesting for the readers of the Journal and can be published in its current form.
Reviewer 2 Report
Comments and Suggestions for Authors
Authors have taken into consideration all the reviewers's comments and they have improved the quality of the paper significantly.
Regards
Comments on the Quality of English LanguageMinor changes